# Health Management: Evaluating the Relationship between Organizational Factors, Psychosocial Risks at Work, Performance Management, and Hospital Outcomes

**DOI:** 10.3390/healthcare11202744

**Published:** 2023-10-16

**Authors:** Tânia Gaspar, Diego Gomez-Baya, Fábio Botelho Guedes, Manuela Faia Correia

**Affiliations:** 1Digital Human-Environment Interaction Labs (HEI-LAB), Universidade Lusófona, 1749-024 Lisbon, Portugal; 2Institute of Environmental Health (ISAMB), Lisbon University, 1649-028 Lisbon, Portugal; fabioguedes_93@hotmail.com (F.B.G.); mcorreia@lis.ulusiada.pt (M.F.C.); 3Department of Social, Developmental and Educational Psychology, Universidad de Huelva, 21004 Huelva, Spain; diego.gomez@dpee.uhu.es; 4COMEGI—Centro de Investigação em Organizações, Mercados e Gestão Industrial, Lusiada University, 1349-001 Lisbon, Portugal

**Keywords:** health organizations, governance, organizational management, human resources management, evaluation

## Abstract

Introduction—Health system (HS) health organizations are complex and are in constant dynamic interaction with multiple elements, including political, environmental, societal, legal, and organizational factors, along with human components, such as human resources, patients, and other stakeholders. Objective—This research aimed to study three HS organizations, identifying and characterizing the elements of health organizations and the factors related to professionals, determining their influence on economic and financial performance results, as well as results related to the professionals and to the patients comprising the institutions. Method—A quantitative study was conducted in which data were collected through questionnaires from various sources to better understand and characterize the factors related to organizations, professionals, and patients (470 health professionals and 768 patients). To test the integrated evaluation model for health organizations, path analysis was used. Results—The results reveal that the organizational culture (OC) presents a positive relationship between the professional’s quality of life (QL) and the performance management (PM) of the professionals, along with a negative relationship with the psychosocial work risks (PWR). There is also a relationship between the OC and patient satisfaction (PS), professional job satisfaction (PJS), and economic and financial results (EFR). In the relationship between the processes and the results, there are significant relationships between PM and PJS and PWR and PJS. In terms of the results, there is a significant relationship between the EFR and the PS. Conclusions—This study contributes to a deeper knowledge of the factors that influence the quality of health organizations and their results and produces recommendations for health organizations to address the current challenges.

## 1. Introduction

### 1.1. General Systems Theory Applied to Healthcare Organizations

General systems theory (GST) is considered a holistic, interdisciplinary approach to science that emphasises the importance of context and environment. A system can be defined as the interaction and/or interdependence of various parts that form the whole, the various parts as a whole serving a common goal and interacting with the environment external to the system [1]. TGS focuses on the relationships between the parts and how they articulate as a whole. The properties of a system are determined by how the parts are organized and how they interact. In this way, a system is a complex, interconnected structure of parts. The whole is different and even greater than the sum of all the parts [2].

Health systems are open complex systems with interrelated subsystems that function in a dynamic equilibrium managed by communication and information [3] and consequently, must be understood from a systemic perspective [4].

The multifaceted nature of health systems and the spread of direct and indirect responsibilities across multiple sectors pose challenges for performance monitoring. In response, in recent years the World Health Organization (WHO) [5] and its partners have been working to reach a broad consensus on key indicators and effective methods and measures for understanding and evaluating health systems, including inputs, processes, and outcomes.

Kaplan et al. [6] identify the principles of applying GST to health organizations. According to the authors, the system must include: (1) a patient-centered system processes, in which the needs and perspectives of the patient should be at the center of the whole process in organizational terms; in terms of planning, implementation, and evaluation; and in terms of the involvement of professionals; (2) the creation of a system of excellence through the identification and establishment of good practices; (3) adjustments to the needs of patients, care, and other circumstances; (4) consideration of a learning process involving scientific evidence and the experience of local change; (5) an emphasis on the interdependence and interconnection between processes and different actors; (6) an emphasis on efficient and effective teamwork, leadership, and coordination; (7) a consideration, but avoidance, of performance errors in the system; (8) the implementation of rewards for elements that contribute to the continuous improvement of the system; (9) encouragement of education/training and research for health professionals, along with partnerships with other relevant scientific areas (e.g., technologies); and (10) the promotion of an organizational culture of leadership and communication that strengthens teamwork and results.

### 1.2. Quality of Healthcare

Promoting quality healthcare is the main objective of health systems. Creating shared value is the goal that should unite all elements of the system. Health systems are dynamic systems in which the patient must be at the center.

The establishment of quality of health services is one of the central goals of health systems. The evaluation of health systems and their performance includes indicators such as access, equity, quality, safety, processes, outcomes, and costs of health services [7].

The quality assessment of health organizations, from a systemic perspective, should consider several key concepts, such as society, strategy, leadership, plans, patients, workers, information and knowledge, people, processes, and outcomes. According to the OECD [8], the process of strategic management towards achieving quality and improved health performance consists of six steps: (1) the diagnosis or analysis of stakeholder influences, the strategic intent of the organization, and analysis of the external and internal environments; (2) the formulation or definition of the plan and strategic options; (3) the implementation and execution of the strategy through the action plan; (4) the control and monitoring of strategic performance through specific indicators; (5) the collection of feedback, i.e., necessary changes and adaptations in the strategic trajectory; and (6) the development of processes and practices that promote organizational learning in strategic management.

### 1.3. Management in Health Services

The World Health Organization [5] suggests the following management functions, such as policy and planning, financial management and planning, human resource management and planning, governance and evaluation, service delivery management and planning, information, and performance management in regards to health services. For quality management of health organizations, quality assurance, accreditation, patient satisfaction and safety, waste management and monitoring, the evaluation of service quality, risk management, the supervisory climate, and organizational support should be considered. At the management level, health organizations require balance in four areas: (1) ensuring an ideal number of managers at all levels of the health system; (2) ensuring that managers have adequate skills; (3) creating a management support system; and (4) creating appropriate conditions in the working environment. These four conditions are closely linked and should each be strengthened.

### 1.4. Organizational Culture

Organizational culture and subcultures, as a system that shares basic assumptions, values, guidelines, principles, and beliefs, define appropriate behavior within an organization in the face of external adaptation and/or internal integration problems, conditioning levels of effectiveness, efficiency, and individual and organizational performance [9,10,11,12,13].

The contrasting values model of [14] allows for the characterization of different organizational cultures using two dimensions combining four quadrants: (1) the flexibility/dynamism-stability/control dimension and (2) the internal orientation/integration-external orientation/differentiation dimension. For the authors the four quadrants of the model—clan culture, hierarchical culture, market culture, and adhocracy culture—emerge as a robust method of explaining the different orientations and values of competition in organizations. Each of these types of organizational culture can be characterized by assumptions, orientations, and values.

The organizational clan culture is the approach most focused on the promotion of the quality of life, well-being, and healthy workspaces of professionals (healthy workplace) and is associated with better job satisfaction, fewer psychosocial risks at work, and consequently, better results, namely better professional performance and customer satisfaction [15]. Consequently, this approach exhibits a greater ability to attract and retain professionals, improving their commitment and enhancing their performance, creativity, and innovation capacity [9,16]. It allows for reducing risk and destabilizing factors at work, such as conflicts, complaints, turnover, absenteeism, performance problems, accidents, and disability [17]. Both the organization and the professionals share the responsibility for maintaining and improving well-being and quality of life.

### 1.5. Well-Being, Quality of Work Life

Well-being at work is associated with psychological well-being; life satisfaction [18], satisfaction and perceived meaning in relation to work; social well-being, related to interpersonal relationships, perceived support, trust, and justice; and physical well-being, associated with health indicators, sense of energy, and disease indicators, such as exhaustion and stress [19,20,21]. This concept is also described in the definition of quality of life proposed by the WHOQoL [22], which includes physical, psychological, social, and environmental health.

High levels of well-being, quality of work life, and job satisfaction are associated with improved performance [23], along with lower levels of turnover [24] (Proudfoot et al., 2009), burnout [25]. Maintaining a professional activity in an unfavorable work environment and exposure to certain psychosocial work risks can affect the quality of life and mental health of professionals [26,27,28,29] and are associated with more negative physical and psychological health indicators. The concept of psychosocial occupational risk was defined by the International Labor Organization (ILO) [30] as risks resulting from the interaction between factors in the work environment of the organization and the occupational risks, with an impact at the level of workers’ overall health and performance. It is important to distinguish between risks and psychosocial factors at work. Risks are management and organizational characteristics intrinsic to the organization which are potentially harmful to the health of the professional. Psychosocial factors are related to the risks to which the professional is exposed and how these risks may be modified by the individual, along with the subjective characteristics of professionals that may act as either protective or risk factors for overall health [11,12,31].

### 1.6. Psychosocial Risks at Work

EU-OSHA [32,33] and the Directorate-General for Health [34,35] describe the main factors that should be considered when assessing psychosocial risks at work, namely organizational culture and leadership relationships, work content, work overload, working hours, level of control and autonomy, interpersonal relationships at work, career development possibilities, work–family relationship, and work environment and equipment.

The Copenhagen psychosocial model is identified as one of the most comprehensive and best suited models for the adoption of occupational health and safety strategies [31,36]. In a study developed by Lega, Prenestini, and Spurgeon [37], it is suggested that the performance of healthcare organizations and the whole healthcare system is correlated with management practices, leadership and organizational culture, and management approaches and values.

### 1.7. Professionals Performance Management

Guest [38] argues that, without losing sight of the organization’s results, the understanding of human resource management should be focused on employee results, not merely on the organization’s performance. Thus, performance management is a key process in the sustainability of organizations. According to Kinicki, Jacobson, Peterson, and Prussia [39], performance management is a set of management processes and behaviors, which aims to define, assess, motivate, and develop the professionals’ performance, with implications at the level of organizational performance. These authors defined six key dimensions for the assessment of performance management: (1) the goal-setting process; (2) communication; (3) feedback; (4) coaching; (5) the provision of consequences; and (6) setting and monitoring expectations linked to performance. The authors add that all these dimensions of performance management are highly correlated with dimensions of leadership and with organizational outcomes and performance.

### 1.8. Promoting Quality in Healthcare Organization Management

A paper developed by Smith, Mossialos, and Papanicola [40] for the World Health Organization’s European Observatory on Health Systems and Policies explores the role of performance evaluation in improving health systems and organizations, concluding that performance evaluation allows policy makers to improve and monitor health systems more effectively. In a study by Braithwaite et al. [41], eight countries were compared in regards to health system performance indicators. The most commonly mentioned indicators were, in the following order: quality and safety, access, patient experience, population health outcomes, and efficiency. The study by Vermeeren, Steijn, Tummers, Lankhaar, Poerstamper, and Beek [42] highlights that a multidimensional perspective on outcomes is especially relevant for health care organizations and concludes that HR practices in health care organizations are directly or indirectly related to the increase in three types of outcomes: financial, organizational, and staff-related. Ott and Dijk [43] add that professional satisfaction with the healthcare organization is directly associated with patient satisfaction. Quality of life (QoL) is associated with the professionals’ well-being and job satisfaction, as well as the interest that the organization has in the potential effect of QoL on productivity and quality.

The organizational culture is significantly related directly and indirectly to QoL, as well as to organizational performance. A positive organizational culture characterized by a work environment based on trust, honesty, and fairness increases professionals’ QoL, job satisfaction, and commitment [44]. Improving the performance of professionals involves the effective implementation of QoL, strengthening the organizational culture in the work environment according to the expectations of professionals, and supporting the goals of the organization [45]. It becomes essential to identify and characterize the factors of the organization and the traits of professionals that influence their quality of life and consequently, influence their performance and other organizational outcomes, as well as minimize the impact of psychosocial risks [15,46,47,48].

The evaluation of healthcare organizations is carried out in a segmented way: organizational factors, psychosocial risks at work, the performance of professionals, and the satisfaction of professionals or patients are evaluated; no model has been identified in the literature that evaluates all of these aspects of the organization simultaneously, enabling the establishment of a relationship between them.

In view of the above, the central objective of this study is to propose a comprehensive and multidimensional model to assess the management and quality of public health organizations, integrating and comparing the different component processes of the system, namely, the impact of organizational (OC), psychosocial (RPT), and professional-related (QL, performance management) factors, illustrating how they relate to outcomes (patient satisfaction, professional satisfaction, and economic and financial performance outcomes).

Based on the established objective, the following research hypotheses are proposed:

**Hypothesis** **1 (H1).**
*A dominant and well-defined organizational culture has a positive influence on the results of public health organizations;*


**Hypothesis** **2 (H2).**
*A high quality of life among health professionals has a positive influence on the results of public health organizations;*


**Hypothesis** **3 (H3).**
*Greater psychosocial risks at work have a negative influence on the results of public health organizations;*


**Hypothesis** **4 (H4).**
*More effective performance management has a positive influence on the results of public health organizations;*


**Hypothesis** **5 (H5).**
*A dominant and well-defined organizational culture is associated with a higher quality of life for health professionals;*


**Hypothesis** **6 (H6).**
*A dominant and well-defined organizational culture is associated with fewer psychosocial risks at work;*


**Hypothesis** **7 (H7).**
*A dominant and well-defined organizational culture is associated with better performance management among healthcare professionals.*


## 2. Method

### 2.1. Participants

The present study involves two samples: a sample consisting of health professionals and a sample consisting of patients from these organizations.

#### 2.1.1. Professionals

The sample of healthcare professionals is composed of 470 participants: 376 females (80%), aged between 20 and 68 years, with a mean of 44.18 and a standard deviation of 9.8. Regarding marital status, most are married or cohabiting (79.9%), and the remaining are single, divorced, or widowed (20.1%). A total of 71.1% of the participants reported having children. Data were collected in three public hospital organizations: 20.8% in Organization A (located in the Central Region of Mainland Portugal); 16.2% in Organization B (located in the Greater Lisbon Region); and 63% in Organization C (located in the Northern Region of Mainland Portugal). A total of 21.5% of the participants mention having a chronic illness. In terms of education, 16.4% of the participants achieved up to the 12th grade of schooling (inclusive), 55.7% had a college degree, and 27.9% had a master’s and/or a PhD.

In regards to the professional group to which they belonged, the most represented group is the nursing profession (38.7%), followed by the medical profession (13.6%). A total of 74.7% of the participants were senior technicians or the equivalent, and 25.3% of the participants were other types of professionals, such as operational technicians.

#### 2.1.2. Patients

The sample of patients from the three health care organizations under study was composed of 768 participants; 463 were female (60.3%), aged between 18 and 68 years. The professionals were divided into four age groups: participants aged between 18 and 24 years (2.3%), participants aged between 25 and 44 years (13.3%), participants aged between 45 and 64 years (41.5%), and participants aged 65 years or more (42.8%). Regarding the employment situation, 2% were students, 38.9% were workers, 8.9% were unemployed, 46% were retired, and 4.3% declared that they were in another situation.

### 2.2. Instruments

The questionnaire administered to the professionals included sociodemographic questions and five scales to evaluate the variables under study: organizational culture, quality of life, psychosocial risks at work, performance management, and satisfaction with work.

#### 2.2.1. Organizational Culture (OC)

A translated and adapted version of the Organizational Culture Assessment Instrument (OCAI), by Cameron and Quinn [9], was used to measure organizational culture. The OCAI constitutes a measure composed of 24 items, which serves to diagnose the type of organizational culture prevailing in the organization, organised into four dimensions with six items each: clan culture (α = 0.82), adhocracy culture (α = 0.85), hierarchy culture (α = 0.80) and market culture (α = 0.69).

The Portuguese version of the scale presented by Cruz and Ferreira [49] was used, employing a 5-point Likert-type scale (1—I strongly disagree; 5—I strongly agree).

#### 2.2.2. Quality of Life (QL)

The World Health Organization Quality of Life (WHOQOL) [22] instrument was used to assess quality of life (QL).

The WHOQOL-BREF instrument comprises 26 questions (question numbers 1 and 2 on general quality of life questions), and the answers employ a Likert-type scale (from 1 to 5; the higher the score, the better the quality of life). Apart from these two questions (1 and 2), the instrument has 24 items that are organized into four dimensions: physical QoL (7 items) (α = 0.87), psychological QoL (6 items) (α = 0.84), social QoL (3 items) (α = 0.64), and environmental QoL (8 items) (α = 0.78). The Portuguese version of the WHOQOL-BREF, translated and adapted by Canavarro et al. [50] and Vaz Serra et al. [51], was used.

#### 2.2.3. Psychosocial Work Factors

The Copenhagen Psychosocial Questionnaire—COPSOQ II (middle version), by Kristensen [52], translated and adapted into Portuguese by Silva et al. [53], was used to measure the psychosocial work factors. The instrument consists of 76 items, organized into 29 dimensions: work demands (6 items), work organization and content (6 items), social relationships and leadership (7 items), work–individual interface 4 items), workplace values (5 items), personality (1 item), health and well-being (7 items), and offending behaviors (4 items); the responses employ a 5-point Likert-type scale (1—never/almost never, to 5—always/almost always). Cronbach’s Alpha values range between 0.20, in the vertical trust dimension, to 0.90, in the leadership quality dimension [53].

#### 2.2.4. Performance Management (PM)

A translated and adapted version of the Performance Management Behavior Questionnaire (PMBQ), by Kinicki et al. [39], was used to measure the performance management of health professionals (self-assessment).

The instrument includes 27 items, organized into 6 dimensions: goal setting process (5 items) (α = 0.91), communication (4 items) (α = 0.86), feedback (5 items) (α = 0.85), coaching (5 items) (α = 0.91), consequence setting (3 items) (α = 0.93), and setting and monitoring performance expectations (5 items) (α = 0.71). The response interval used a 5-point scale (1—rarely or never; 5—often or always).

#### 2.2.5. Professional Job Satisfaction (PJS)

A translated and adapted version of the Satisfaction of Employees in Health Care (SEHC) scale [54] was used to assess job satisfaction.

The instrument was translated and adapted through the following translation and back-translation procedure: translation of the English version into Portuguese by two researchers, comparison and homogenization of the versions, translation of the resulting version into English, comparison of this version with the original version by an expert in English, and confirmation of the final version, with the agreement of both researchers [55]. The scale consists of 20 items, 18 of them with a 4-point response scale (1—strongly disagree, to 4—strongly agree), and these items were organized into three dimensions: relationship with management and supervisors (11 items) (α = 0.89), work content (5 items) (α = 0.70), and relationship with co-workers (2 items) (α = 0.70). The scale also includes two additional items: “I would recommend this service to others as a good place to work”, with a 4-point response scale (1—not at all, to 4—yes, definitely) and “How would you rate this health facility as a place to work on a scale from 1 (the worst) to 10 (the best)?”.

#### 2.2.6. Patient Satisfaction (PS)

The Questionnaire on Health System User Satisfaction (QSUSS) was used to measure patient satisfaction [34,35].

The following question was used for the present study: “In your opinion, how does the Portuguese Health System work?” with the following response hypotheses: “Works well”; “Needs minor changes/adjustments”; “Needs major changes/adjustments”; “Needs to be completely restructured”.

#### 2.2.7. Economic and Financial Results (EFR)

To evaluate the economic and financial results of the organization, a standardized indicator of operating expenses reported by the Central Administration of the Health System (ACSS) was used, with a value that varies between 1 and 2—a higher value refers to higher operating expenses (https://benchmarking-acss.min-saude.pt/BH_AcessoDashboard, accessed on 3 March 2020).

### 2.3. Procedure

The data collection procedure included different phases. In the first phase, the study was submitted to the Ethics Committee of the Lisbon Academic Medicine Center of the Lisbon North Lisbon Hospital Center of the Faculty of Medicine of the University of Lisbon and obtained a favorable opinion.

For the implementation of the research study, after the identification of the hospitals that would be the target of the study and the approval of the respective administrations, meetings were held with the clinical directors of the specialties involved for the presentation of the project and involvement in the data collection process [56]. After the project was presented to the administration and collaborators of the hospitals under study, and after their agreement to participate in the study was obtained, the project was submitted to the ethics committees and the boards of directors of the three participating hospitals.

After the collection of all the necessary authorizations, we moved on to data collection. The selection of hospitals was carried out by the Ministry of Health’s Directorate-General for Health, which selected three national hospitals specializing in a particular area in the three regions of the country (North, Center, and South).

The quantitative instrument was applied through a link which was disclosed to the participants. The patient satisfaction questionnaire was administered through telephone interviews. The patients were contacted by telephone by professionals from the respective hospital. The professionals were trained by the research team and followed a provided script. The telephone interview initially contained information regarding informed consent, and if the patient agreed to take part, the professional began by asking questions related to satisfaction with healthcare.

In regards to the sample of professionals, all the professionals in each hospital were invited to take part in the study. The administration of each hospital sent all the professionals a link to the informed consent form and the study questionnaire via their professional email addresses. The participation of at least 5% of the professionals in each hospital was accepted. With regard to patients, a random selection was made of those who had received treatment in the hospital in the last 6 months, and informed consent was obtained and the questionnaire was administered through telephone interviews.

The model included data from three different sources: from professionals (N = 470), from patients (N = 768), and from economic and financial outcomes (https://benchmarking-acss.min-saude.pt/BH_AcessoDashboard, accessed on 3 March 2020).

The information was collected for each of the hospitals, and the total database included, in the case of professionals, the results per subject, and in the case of patients and economic–financial outcomes, the average value of the results for each of the hospitals.

The patients were not paired with professionals, and the cluster analyzed was separate for each hospital, the total results of professionals for each of the variables, the total results of the hospital for patient satisfaction, and the total economic and financial results.

For each of the instruments, anonymity and confidentiality are ensured, since the researcher did not have cumulative access to the participant’s identification or the collected data. The association with the data was always achieved by an identification number. For the study of the model, the results of the instruments completed by the professionals, financial economic performance results, and patient satisfaction evaluations were integrated.

### 2.4. Data Analysis and Treatment

For the analysis of quantitative data, such as those obtained through the application of questionnaires, statistical analysis was performed using the SPSS software, version 24.0. Confirmatory factor analyses and structural equation models were performed using EQS software, version 6.3.

Data cleaning was used to check for incorrect values; for example, ages under 18 or over 110 (these were not identified). As the questionnaire could only be submitted with all the questions answered, no missing values were identified.

The original instrument, which included the complete scales previously mentioned, consisted of a total of 241 items, with an average completion time of 30 min. In order to propose a user-friendly instrument for effective and sustainable use in the assessment and monitoring of the management and quality of health care organizations, a process of item reduction was conducted for each of the scales under study, and a path analysis was performed to determine the quality and robustness of the proposed model. In the item reduction process, the following analyses were conducted for each of the instruments used in the study (i.e., organizational culture assessment instruments, quality of life assessment instruments, psychosocial risks at work assessment instrument, performance management assessment instrument, and job satisfaction assessment instrument): first, exploratory factor analysis SPSS; second, factor reliability analysis SPSS; third, Pearson correlation analysis SPSS; fourth, confirmatory factor analysis of all EQS scales; and fifth, structural equation model EQS.

For the confirmatory factor analysis, structural equation modeling was performed. The ratio between the Chi-square (χ^2^) and the degrees of freedom (gl), and the fit indices NNFI (non-normed fit index), CFI (comparative fit index), RMSEA (root mean square error of approximation), and interval were considered. The χ^2^ indicates the magnitude of the discrepancy between the observed and the modeled covariance matrix, assessing the probability of the model’s fit to the data. Its ratio in relation to the degrees of freedom (χ^2^/gl) is generally considered, whose appropriate values are between 1 and 3 [57,58]. The NNFI and CFI indices calculate the relative fit of the observed model by comparing it with a base model, with values above 0.95 indicating optimal fit, and those above 0.90 indicating adequate fit [59,60]. The RMSEA is a measure of discrepancy, with results up to 0.06 being considered good, but those up to 0.08 deemed acceptable [61,62].

## 3. Results

With the purpose of testing the multidimensional model for assessing and monitoring the quality of health organization management, based on the instruments used in the study, the items of each instrument were reduced through a rigorous procedure that took into account the theoretical and conceptual knowledge of the variables and the quality and robustness of the statistical analysis and treatment results.

### 3.1. Item Reduction Process

In the first phase, for item reduction, the following analyses were performed for each of the instruments used in the study: organizational culture assessment instruments, quality of life assessment instruments, psychosocial risks at work assessment instrument, performance assessment instrument, and job satisfaction assessment instrument. The analyses were performed as follows: first, exploratory factor analysis (SPSS Statistics 24); second, factor reliability analysis (SPSS 24); third, Pearson correlation analysis (SPSS Statistics 24); fourth, confirmatory factor analysis of all EQS scales; and fifth, the structural equation model (EQS 6.3).

The item reduction process for each instrument used followed the following steps: (1) sensitivity study; (2) exploratory factor analysis with the use of direct oblimin rotation, with coefficients > 0.70 selected and extraction, based on eigenvalue; (3) exploratory factor analysis, and (4) internal consistency analysis (reliability) for the one-dimensional scale Cronbach’s alpha.

To test the quality of the reduced scale models, confirmatory factor analyses were performed using the EQS program.

#### 3.1.1. Organizational Culture

From the item reduction process of the organizational culture assessment scale, there resulted a factor that explained 35.35% of the variance, and which was considerably far from the second factor, with only 10% explained variance of the EFA. It was decided to select the items with a weight >0.70 of the first factor. The EFA (3) with the eight items, selected in a unidimensional scale total, explained the variance of 62.33, with an eigenvalue of 4.99, a KMO value of 0.90, and (4) a Cronbach’s alpha of 0.91, which does not improve with the exclusion of any of the items.

For the reduced organizational culture scale, the initial model found through confirmatory factor analysis points to a poorly adjusted model, χ^2^(20) = 226.12, *p* = 0.001, NCFI = 0.85, CFI = 0.90, RMSEA = 0.15, and 90% CI RMSEA = 0.13, 0.17. After the recommended associations were integrated with the Lagrange multiplier test, a more robust model was achieved, with χ^2^(17) = 70.13, *p* = 0.001, NCFI = 0.96, CFI = 0.97, RMSEA = 0.08, and 90% CI RMSEA = 0.06, 0.10.

#### 3.1.2. Quality of Life

Regarding the quality of life assessment scale, we aimed to propose a unidimensional scale for quality of life.

The item reduction process resulted in a factor explaining 37% of the variance and considerably far from the second factor, with only 7% explained variance of the EFA. It was decided to select the five items with a weight >0.70 of this first factor. The EFA, with extraction with a fixed number of values (1), with the five selected items, resulted in a factor explaining about 60% of the variance (eigenvalue of 2.99 and explained variance of 59.87%) and a KMO value of 0.79; (4) for the unidimensional scale, the Cronbach’s alpha was 0.83, which did not improve with the exclusion of any item.

Regarding the reduced five-item quality of life scale, the initial model found through confirmatory factor analysis points to a poorly adjusted model, χ^2^(5) = 59.56, *p* = 0.001, NCFI = 0.79, CFI = 0.89, RMSEA = 0.15, and 90% CI RMSEA = 0.12, 0.19. After the recommended associations were integrated using the Lagrange multiplier test, a more robust model was achieved, with χ^2^(3) = 6.72, *p* = 0.001, NCFI = 0.98, CFI = 0.99, RMSEA = 0.05, and 90% CI RMSEA = 0.00, 0.10.

#### 3.1.3. Psychosocial Rick Factors of Work

The process of item reduction of the EFA psychosocial risk factors of work rating scale revealed 17 factors, with the selected items having more weight in each of the 2 main factors, with the 15 selected items resulting in 2 factors explaining approximately 63% of the variance; the internal consistency analysis (reliability) for each of the two dimensions obtained good results, with a Cronbach’s alpha of 0.92, which does not improve with the exclusion of any of the items, and explains 63% of the variance, with an eigenvalue of 7.21 and an explained variance of 48.08% for factor 1, and an eigenvalue of 2.21 and an explained variance of 14.71% for factor 2. With KMO value of 0.78 for the dimension related to PRW in the area of mental health and 0.94 in the dimension related to Psychosocial Factors at Work at the level of leadership and work.

Regarding the reduced 11-item psychosocial risk factors at work scale—leadership and work content, the initial model found through confirmatory factor analysis points to a poorly adjusted model, χ^2^(44) = 278.49, *p* = 0.001, NCFI = 0.89, CFI = 0.92, RMSEA = 0.15, and 90% CI RMSEA = 0.10, 0.12. After the recommended associations were integrated by the Lagrange multiplier test, a more robust model was achieved, with χ^2^(41) = 278.49, *p* = 0.001, NCFI = 0.97, CFI = 0.98, RMSEA = 0.06, and 90% CI RMSEA = 0.05, 0.07.

Regarding the reduced 4-item psychosocial risk factors of work—mental health scale, the initial model found through confirmatory factor analysis points to a poorly adjusted model, with χ^2^(2) = 13.43, *p* = 0.001, NCFI = 0.95, CFI = 0.98, RMSEA = 0.11, and 90% CI RMSEA = 0.06, 0.17. After the recommended associations were integrated using the Lagrange multiplier test, a more robust model was achieved, with χ^2^(1) = 0.28, *p* = 0.001, NCFI = 1.00, CFI = 1.00, RMSEA = 0.00, and 90% CI RMSEA = 0.00, 0.10.

#### 3.1.4. Performance Management

With regard to the reduced version of the performance management assessment scale, we aimed to propose a unidimensional scale of performance.

The item reduction process resulted in a factor that explained 35% of the variance and which was considerably distant from the second factor, with only 6% of explained variance of the EFA; it was decided to select the five items with weight >0.70 of this first factor; exploratory factor analysis, with extraction with a fixed number of values (1), with the five selected items, resulted in a factor explaining approximately 73% of the variance (eigenvalue of 3.67 and explained variance of 73.48%) and a KMO value of 0.87; in regards to the internal consistency analysis (reliability) for the unidimensional scale, the Cronbach’s alpha was 0.91, which did not improve with the exclusion of any item.

Regarding the reduced five-item performance management scale, the initial model found through confirmatory factor analysis points to an adjusted model, with χ^2^(5) = 18.36, *p* = 0.001, NCFI = 0.95, CFI = 0.97, RMSEA = 0.08, and 90% CI RMSEA = 0.05, 0.13. After the recommended associations were integrated using the Lagrange multiplier test, a more robust model was achieved, with χ^2^(4) = 7.87, *p* = 0.001, NCFI = 0.98, CFI = 0.99, RMSEA = 0.05, and 90% CI RMSEA = 0.00, 0.09.

#### 3.1.5. Professional Satisfaction

Regarding the scale of evaluation of professional satisfaction with work, we aimed to propose a unidimensional scale for professional satisfaction with work.

The process of item reduction resulted in four factors, with 62% total variance explained. We chose to select the items of the first factor, resulting in six items; exploratory factor analysis, with extraction with a fixed number of values (1), with the six selected items results in a factor that explains about 58% of the variance (eigenvalue of 3.49 and explained variance of 58.11%) and a KMO value of 0.87; in regards to the analysis of internal consistency (reliability) for the unidimensional scale, Cronbach’s alpha was 0.86, which did not improve with the exclusion of any items.

Regarding the six-item reduced scale of professionals’ job satisfaction, the initial model found through confirmatory factor analysis points to an adjusted model, with χ^2^(9) = 35.12, *p* = 0.001, NCFI = 0.95, CFI = 0.97, RMSEA = 0.08, and 90% CI RMSEA = 0.05, 0.11. After the recommended associations were integrated by the Lagrange multiplier test, a more robust model was achieved, with χ^2^(8) = 20.56, *p* = 0.001, NCFI = 0.97; CFI = 0.99, RMSEA = 0.06, and 90% CI RMSEA = 0.03, 0.09.

Table 1 shows that the values obtained for each of the variables under study presenting positive and moderate mean values. The organizational culture presents a mean value of 3.05 (SD = 0.84), the quality of life presents a higher mean value of 3.90 (SD = 0.63), the psychosocial factors present moderate mean values, the performance management presents a mean value of 3.72 (SD = 0.58), and the satisfaction of professionals and patients present moderate mean values of 2.32 (SD = 0.58) and 1.82 (SD = 0.07), respectively. The same was true for the mean value of financial economics—operating expenses, which was 1.17 (SD = 0.06).

Table 2 describes the correlations between the variables under study and Cronbach’s alphas. The vast majority of the variables are correlated in a statistically significant way. We highlight the negative correlation between organizational culture and psychosocial factors related to content and leadership (r = −0.61; *p* < 0.001), and the positive correlation between organizational culture and job satisfaction (r = 0.67; *p* < 0.001); the negative correlation between psychosocial risks related to content and leadership and job satisfaction (r = 0.75; *p* < 0.001); the negative correlation between psychosocial risks related to mental health and the quality of life of professionals (r = −0.60; *p* < 0.001); and the negative correlation between patient satisfaction and poor financial economic performance results (r = −0.97; *p* < 0.001). It is confirmed that all scales under study show good internal consistency values, ranging from a Cronbach’s alpha of 0.93 in regards to the psychosocial factors related to content and leadership scale, and a Cronbach’s alpha of 0.83 in regards to the quality of life and psychosocial risks related to mental health scales.

A path analysis was performed to study the Integrated and Multidimensional Model for the Assessment of the Quality of Management of Health Organizations, and the initial model found proved to be a poorly adjusted model (Table 3). After integrating the associations recommended by the Lagrange multiplier test and removing the links indicated through the Wald test, a more robust model was achieved (Figure 1 and Table 3).

In the relationship between organizational culture (input) and processes, there is a significant positive relationship between organizational culture and quality of life of professionals (β = 0.22), and between organizational culture and professionals’ performance management (β = 0.15), as well as a negative relationship between psychosocial work risks at the level of work and leadership characteristics (β = −0.61) and psychosocial work risks at the level of mental health (β = −0.30).

In the relationship between organizational culture (input) and the results, we found a positive correlation with patient satisfaction (β = 0.22) and professional satisfaction (β = 0.35) and a negative relationship with poor economic and financial results (β = −0.25).

At the processes level, significant negative relationships were found between quality of life and psychosocial work risks at the work and leadership characteristics level (β = −0.26), between quality of life and psychosocial work risks at the mental health level (β = −0.57), and between psychosocial work risks at the work and leadership characteristics level and psychosocial work risks at the mental health level (β = 0.25). There are also significant negative relationships between performance management and psychosocial work risks at the work and leadership characteristics level (β = −0.12).

In the relationship between processes and outcomes, significant relationships were found between performance management and job satisfaction (β = 0.09) and between psychosocial work risks in work and leadership characteristics and job satisfaction (β = −0.51).

In terms of results, there is a significant relationship between economic and financial results and patient satisfaction with the National Health System (β = −0.97).

## 4. Discussion

Next, the results will be discussed in an integrated way in light of the general systems theory, reflecting the relationship between the different systems associated with healthcare organizations, namely the strength and meaning of these relationships.

These results provide a direct answer to the study hypotheses.

H1—A dominant and well-defined organizational culture has a positive influence on the results of public health organizations. The results confirm statistically significant correlations (*p* > 0.05) between the types of organizational culture and the results under analysis (staff satisfaction, patient satisfaction, and economic and financial performance results), highlighting the strong correlation between clan culture and staff satisfaction (R = 0.67).

H2—High quality of life among health professionals has a positive influence on the results of public health organizations. The results partially confirm this hypothesis. There is a moderate and statistically significant correlation (R = 0.31) between professionals’ quality of life and job satisfaction.

H3—Greater psychosocial risks at work have a negative influence on the results of public health organizations. The results confirm this hypothesis. There is a strong and statistically significant negative correlation (R = −0.75) between psychosocial risks at work and job satisfaction. The relationship is statistically significant, but weak in relation to patient satisfaction and economic and financial performance results.

H4—Effective performance management has a positive influence on the results of public health organizations. The results partially confirm this hypothesis. There is a strong and statistically significant correlation (R = 0.35) between performance management and job satisfaction. The relationship is not confirmed in relation to patient satisfaction and economic and financial performance results.

H5—A dominant and well-defined organizational culture is associated with a higher quality of life for healthcare professionals. Given this hypothesis, we found that quality of life is positively and statistically significantly correlated with organizational culture (R = 0.22).

H6—A dominant and well-defined organizational culture is associated with fewer psychosocial risks at work. In view of this hypothesis, we found that psychosocial risks at work are strongly negatively and statistically significantly correlated with organizational culture (R = −0.61).

H7—A dominant and well-defined organizational culture is associated with better performance management among healthcare professionals. Given this hypothesis, we found that performance management is positively and statistically significantly correlated with organizational culture (R = 0.15).

Healthcare organizations face a number of challenges in terms of the results they set out to achieve. Health and well-being, as well as professional and patient satisfaction, will influence economic and financial results [19,20,21,63]. Psychosocial risk factors can be individual factors, organizational factors, and the interaction between them, as well as the level of exposure to risks and the concrete impact this exposure has on the professional’s general health. Individual characteristics include psychological factors such as coping style, personality and cognitive characteristics, along with sociodemographic characteristics, such as gender, age, marital status, and physical and mental health history. EU-OSHA [26,32] and the Directorate-General for Health [34,35] describe the main factors that should be considered when assessing psychosocial risks at work: the organization’s culture and leadership relationships, work content, work overload, working hours, level of control and autonomy, interpersonal relationships at work, career development possibilities, work–family relationships, and work environment and equipment. The impact of psychosocial risks on the health and performance of professionals can have chronic and long-term consequences [19,20,21], in terms of their physical health, mental health, chronic stress, and absenteeism from work [12].

The model obtained can be understood in the light of the general system theory, i.e., the interrelationship and influence between the variables makes it possible to identify and understand factors exhibited by the organization, the professionals, the results, and their relationship, influencing the quality of life of the professionals, as well as the performance and the results of the organization [46,64,65].

The OECD [8] and Clarkson et al. [66] argue that assessing the quality of healthcare organizations from a systemic perspective must take into account various fundamental concepts, such as leadership, strategy, plans, patients, society, information and knowledge, people, processes, and outcomes.

The model integrates variables of the organization (organizational culture), of the professionals (quality of life), of the professionals in the organization (psychosocial risks at work and performance management), and results (professional satisfaction, patient satisfaction, and economic and financial performance results).

It is confirmed that a well-defined organizational culture has a positive influence on the results of the analyzed health organizations of the public subsystem. The results confirm statistically significant correlations between the types of organizational culture and the outcomes under analysis (staff satisfaction, patient satisfaction, and economic and financial performance results). Higher psychosocial risks at work are found to have a negative influence on organizational outcomes. A strong and statistically significant negative correlation was identified between psychosocial risks at work and job satisfaction. Effective performance management has a positive influence on the outcomes of the healthcare organizations studied, with a strong and statistically significant correlation between performance management and job satisfaction. A well-defined organizational culture is associated with higher health professional quality of life, fewer psychosocial risks at work, and better performance management of health professionals.

The model obtained can be understood in the light of the general system theory, i.e., the interrelationship and influence between the variables allows for the identification and determination of organizational, professional, and outcome factors and their respective relationships that influence the quality of life of professionals, as well as the performance and the results of the organization [10,12,46,47,64,65,67].

In the relationship between organizational culture (Input) and results, we found a positive relationship with patient satisfaction and professional satisfaction and a negative relationship with poor economic and financial results.

At the processes level, negative relationships were found between quality of life and psychosocial risks of work in terms of work and leadership characteristics, between quality of life and psychosocial risks of work in terms of mental health, and between psychosocial risks of work in terms of work and leadership characteristics and psychosocial risks of work in terms of mental health. Negative relationships were also found between performance management and psychosocial work risks at the work and leadership characteristics level. In the relationship between processes and outcomes, there are relationships between performance management and job satisfaction and psychosocial work risks in work and leadership characteristics and job satisfaction. In terms of results, there is a relationship between economic and financial results and patient satisfaction with the National Health System.

Health organizations are faced with several challenges in view of the results they propose to achieve. Health and well-being, as well as staff and patient satisfaction, will influence economic and financial outcomes [19,20,21,63,68,69]. Psychosocial risk factors can be individual factors, organizational factors, and the interaction between them, as well as the level of exposure to risks and the concrete impact that risk has on the general health of the professional [70]. As individual characteristics, psychological factors, such as coping style, personality and cognition characteristics; and sociodemographic characteristics, such as gender, age, marital status, and medical history, along with the level of physical and mental health, as also noted [71,72,73]. EU-OSHA [32,33] and Direção-Geral da Saúde [34,35] describe the main factors that should be considered in the assessment of psychosocial risks at work: the culture of the organization and leadership relationships, work content, work overload, working hours, level of control and autonomy, interpersonal relationships at work, possibilities for career development, work–family relationship, and work environment and equipment. The impact of psychosocial risks on the health and performance of professionals may have chronic and long-term consequences [19,20,21] in regards to their physical health [74,75], mental health, chronic stress [11,76], and work absenteeism [77].

The results prompt the conclusion that a health care organization with an organizational culture in which professionals have better quality of life, fewer psychosocial risks at work, and better performance management consequently presents better results in terms of professional and patient satisfaction and better economic and financial performance, demonstrating consistency between the inputs, processes, and results. The valorization of psychosocial risks at work by the organization and management [5] will have an impact on the physical, mental, and social health of professionals [78,79] and on the organization’s health indices related to absenteeism, productivity, professional satisfaction, and turnover [80,81,82].

The OECD [8] and Clarkson et al. [66] argue that the quality assessment of health organizations, from a systemic perspective, should take into account several key concepts, such as leadership, strategy, plans, patients, society, information and knowledge, people, processes, and outcomes.

A strong focus on management, as a key element for global health, may open new and sustainable ways to improve health systems performance [4,11,12,83,84].

This study contributes to a more in-depth understanding of the factors that influence the quality of healthcare organizations and their results, and produces recommendations for healthcare organizations to meet these current challenges. The study used a systemic, multidimensional, and integrative perspective to characterize and understand the factors, actors, and their respective relationships and influences in SNS health organizations, at the political, organizational, human resource, professional, patient, and economic–financial levels. The study used a mixed methodology, using quantitative and qualitative methods, integrating the information collected through various methods and multi-informants, making it more robust.

The difficulties faced by health researchers in terms of data collection are well known. The fact that this research was successful in this respect can be explained because the administrations of the healthcare organizations were involved in the study from the outset; there was awareness-raising among management, middle management, and professionals; and finally, because the results and products of the study can have a direct impact on the assessment and monitoring of the quality and integrated quality and safety systems of the respective healthcare organizations.

The study involved participants from different levels of healthcare organizations, which made it possible to integrate and triangulate the information and thus understand the perception of each of the stakeholders, allowing points of agreement, as well as discrepancies, to be identified.

As a result, we suggest a comprehensive diagnostic model of the factors that influence the results in healthcare organizations, providing greater knowledge of the systems and the relationships between the systems, which can be used to support decision-making, planning, and the implementation of improvements in NHS healthcare organizations. For the regular evaluation and monitoring of SNS health organizations, an integrated evaluation model and method was studied and proposed to assess the impact of implementing improvement measures and consequently, supporting an evidence-based governance process. In addition to the evaluation and monitoring tools, recommendations and suggestions for action for the continuous and sustainable improvement of health organizations in the National Health System are presented and discussed.

We would like to highlight some implications for practice based on the results, recommendations, and products of this study: the importance and impact of organizational culture on processes and results was demonstrated; at the level of professionals, the relevance of involving all professionals was shown, groups with specific needs and levels were identified, and the relationship of QoL and psychosocial risks at work with other processes and results in healthcare organizations was proven; at the level of patients, there was general satisfaction with the SNS, especially with the organizations under study; however, waiting times and involvement in decision making in relation to their health are aspects that need improvement, according to patients’ perceptions.

The main theoretical contribution of this work is the in-depth study of general systems theory in healthcare organizations. Systems and subsystems in healthcare organizations were identified, selected, and studied, and the types and strength of the influences between the different systems and subsystems were analyzed. The study makes it possible to verify the profile of a healthcare organization that obtained more positive indicators for all factors and all levels of analysis. Another important contribution is establishment of the relationship between organizational culture, psychosocial risks at work, and results, as well as the relationship between patient satisfaction and economic and financial results.

## 5. Conclusions

As a product, we present a comprehensive diagnostic model of the factors influencing the outcomes in health organizations which allows for a greater knowledge of the systems and relationships between systems and that can support decision making, planning, and the implementation of improvements in NHS health organizations.

The resulting instrument is user-friendly and allows for the regular assessment and monitoring of NHS health organizations.

The study allows for the identification of a health organization profile through the assessment of different indicators for all factors and all levels of analysis. Another revealing contribution is the relationship established between organizational culture, psychosocial work risks, and results. Moreover, the relationship between patient satisfaction results and economic and financial results is also revealed.

The main limitations of the study were the low level of participation by health professionals in completing the quantitative data collection questionnaire, even though methodological measures were taken to increase participation, such as involving hospital administration and clinical management, internal presentation of the study in a plenary session, and reinforcement for participation. This low level of participation may be due to the length of the questionnaire, which took around 30 min to complete; the low expectations of professionals regarding the impact of their participation, as well as their relationship with the administration and management. There was a much higher response rate in Organization C, which reflected better overall outcomes in terms of quality and results. The fact that the total number of participants was lower than that expected created limitations in terms of analyzing and processing the data in the reduced model proposal. Consequently, this research proposes a smaller instrument (average completion time of 10 min) to increase the participation rate in future research, or even to make it more feasible to use the model in evaluation and monitoring, in the form of a diagnostic tool for use by healthcare organizations.

The sample was completed at the convenience of the responders and included only public health organizations, which limits the generalizability of the results to global health organizations. It would be desirable to continue this research by applying the methodology to a random and representative sample of NHS health organizations, public health organizations with private management, and private health organizations.

We conclude that there are several barriers related to management, health professionals, patients, and economic and financial issues in the process of seeking better quality and performance in regards to health organizations. According to WHO (WHO, 2007) the strengthening of health systems involves the integration, understanding, and management of the main barriers, as well as the good practices, in each of the areas and subsystems of health systems. Kaplan et al. (2013) consider that a health organization is made up of subsystems, resources, and people who have the common goal of improving health. To this end, they argue that processes should be patient-centered in terms of the management of the planning, implementation, and assessment process, and that an organizational culture focused on positive leadership and quality communication that strengthens interpersonal relationships and teamwork should be promoted. Investment in training and research is also fundamental in order to allow management and clinical decisions to be based on scientific evidence.

This paper is part of a wider study on the global evaluation of healthcare in Portugal. As a result of this study, evaluation reports, recommendations, and action plans have been drawn up for each of the hospitals involved, and annual monitoring of the impact of continuous quality improvement will be carried out. The aim is to extend the evaluation and monitoring to other hospitals and primary healthcare organizations (health centers), always involving the different stakeholders, administrations, professionals, and patients.

The Portuguese health system, the NHS, and the basic health law [85,86] present principles and structures that fit into the systems theory and allow for the understanding, assessment, and improvement of health systems at the management, quality, and performance levels. From this perspective, for a better understanding of a complex and multidimensional phenomenon, such as the functioning of Portuguese health organizations, the following factors should be included: organizational factors, professional factors, and patient factors, along with social, cultural, economic, and political factors [87].

The following are the main recommendations for managers of NHS health organizations: (a) to periodically assess and monitor the quality and performance of health organizations from a systemic and integrative perspective (integrating inputs, processes, and outcomes); (b) to promote healthy workplaces by improving psychosocial working conditions, and by promoting the physical, social, and mental health of professionals, for instance by providing them with more availability for non-care activities, such as management, research, and training.

## Figures and Tables

**Figure 1 healthcare-11-02744-f001:**
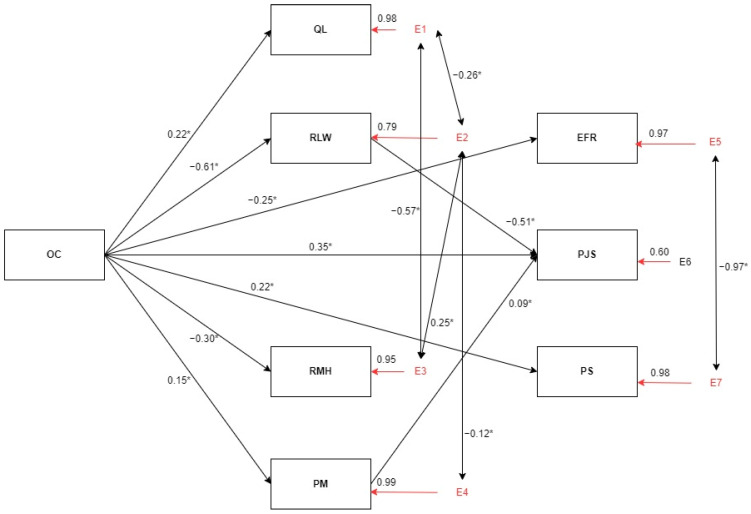
Global health management model—final model. OC = organizational culture; QL = quality of life; RLW = psychosocial risk factors at work—leadership and work content; RMH = psychosocial risk factors at work—mental health; PM = performance management; EFR = economic and financial results; PJS = professionals’ job satisfaction; PS = patient’s satisfaction. * *p* < 0.001.

**Table 1 healthcare-11-02744-t001:** Descriptive analyses of the reduced scales to be included in the model.

	Min	Max	M	SD
Organizational Culture	1	5	3.05	0.84
Quality of Life	1	5	3.90	0.63
Psychosocial Risk Factors at Work—Leadership and Work Content	1	5	2.89	0.84
Psychosocial Risk Factors at Work—Mental Health	1	5	2.85	0.85
Performance Management	1	5	3.72	0.58
Professionals’ Job Satisfaction	1	4	2.32	0.58
Patients Satisfaction	1	4	1.82	0.07
Economic and Financial Results	1	2	1.03	0.06

**Table 2 healthcare-11-02744-t002:** Pearson’s correlations of the model variables and internal consistency.

	1	2	3	4	5	6	7	8
1. Organizational Culture	(0.91) ^#^							
2. Quality of Life	0.22 ***	(0.83) ^#^						
3. Psychosocial Risk Factors at Work—Leadership and Work Content	−0.61 ***	−0.34 ***	(0.93) ^#^					
4. Psychosocial Risk Factors at Work—Mental Health	−0.30 ***	−0.60 ***	0.37 ***	(0.83) ^#^				
5. Performance Management	0.15 **	0.14 **	−0.20 ***	−0.11 *	(0.91) ^#^			
6. Economic and Financial Results	−0.25 ***	0.02	0.11 *	0.01	−0.01	(1)		
7. Patients Satisfaction	0.22 ***	−0.04	−0.10 *	0.01	−0.01	−0.97 ***	(1)	
8. Professionals’ Job Satisfaction	0.67 ***	0.31 ***	−0.75 ***	−0.35 ***	0.25 ***	−0.16 **	0.14 **	(0.86) ^#^

*** *p* < 0.001, ** *p* < 0.01, * *p* < 0.05. ^#^ alpha de Cronbach.

**Table 3 healthcare-11-02744-t003:** Path analyses—Suitability indices of the Integrated and Multidimensional Model for the Assessment of the Quality of Management of Healthcare Organizations.

	χ^2^	gl	χ^2^/gl	NNFI	CFI	RMSEA	Range
Initial Model	1120.23	9	124.47	−0.83	0.41	0.51	0.49–0.54
Final Model	15.22	14	1.09	0.99	0.99	0.014	0.00–0.05

## Data Availability

Data is unavailable due to privacy or ethical restrictions.

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
