# Peer review of "Health Management: Evaluating the Relationship between Organizational Factors, Psychosocial Risks at Work, Performance Management, and Hospital Outcomes"

_healthcare, 2023, doi:10.3390/healthcare11202744_

Round 1
Reviewer 1 Report
The manuscript needs extensive revision because of the lack of logical flow and organization of the contents. Specific comments are given below;
1. Rewrite the abstract in a structured form. It will be more impactful for readers
2. Organize the introduction section with subheadings
3. Write the statement of the study, which must include the research gap and objective of the study
4. I’m unable to find the theoretical framework. Kindly add a relevant theoretical framework as the foundation of the study
5. The present study involves two samples: a sample of health professionals and a sample of patients from these organizations. How did both samples select? Under which sampling technique? What’s the criterion for health professionals and patients to participate in the study? Write it clearly
6. How did the hospitals select? What were the criteria for hospital selection?
7. Provide the revised article with the complete data collection instrument (Questionnaire) for both health professionals and patients. I would like to see it. It should be in English and attached supplementary file with the revised article
8. Is there any data quality check on the collected data? If yes, then explain the procedure of the data quality check
9. What are the limitations and implications of the study? Write in detail
10. What’s the future scope of the study? Write in detail under the context of health professionals and patients
Send me revised article again for review
Author Response
Revisions Paper
“Health Management: evaluating the relationship between or-ganizational factors, psychosocial risks at work, performance management and hospitals outcomes.”
Dear Editor and Reviewer
I greatly appreciate the careful analysis and recommendations indicated by the reviewers. I have made the requested changes; in this document, following the reviewer's recommendation, is the answer that was included in the manuscript, I indicate in track changes. I made substantial changes that greatly improved the paper, thank you very much. If you have any further questions or if needed any additional changes, please just let me know best regards Tania
The manuscript needs extensive revision because of the lack of logical flow and organization of the contents. Specific comments are given below;
- Rewrite the abstract in a structured form. It will be more impactful for readers
The abstract was organised into the following topics: introduction, objective, method, results and conclusions.
- Organize the introduction section with subheadings
The introduction section was organized in the following subheadings
General Systems Theory applied to Healthcare organizations
Quality of healthcare
Management in Health services
Organisational culture
Well-being, work quality of life
Psychosocial risks at work
Professionals Performance
Promoting quality in healthcare organizations management
- Write the statement of the study, which must include the research gap and objective of the study
The evaluation of healthcare organisations is carried out in a segmented way: organisational factors are evaluated, or psychosocial risks at work, or the performance of professionals, or the satisfaction of professionals or patients; no model has been identified in the literature that evaluates all these aspects of the organisation at the same time and that makes it possible to establish the relationship between them.
In view of the above, the central objective of this study is to propose a comprehensive and multidimensional model to assess the management and quality of public health organisations, which integrates and relates different component processes of the system, namely, the impact of organisational (OC), psychosocial (RPT), professional-related (QL, performance management) factors, and illustrates how they relate to outcomes (patient satisfaction, professional satisfaction and economic and financial performance outcomes).
- I’m unable to find the theoretical framework. Kindly add a relevant theoretical framework as the foundation of the study
General Systems Theory (GST) is considered a holistic, interdisciplinary approach to science that emphasises the importance of context and environment. A system can be defined as the interaction and/or interdependence of various parts that form the whole, the various parts as a whole serving a common goal and interacting with the environment external to the system (Cordon, 2013). TGS focuses on the relationships between the parts and how they articulate as a whole. The properties of a system are determined by how the parts are organised and interact. In this way, a system is a complex, interconnected structure of parts. The whole is different and even greater than the sum of all the parts (Chikere & Nwoka, 2015).
Health systems are open complex systems with interrelated subsystems that function in a dynamic equilibrium managed by communication and information (Chuang & Inder, 2009) and consequently must be understood from a systemic perspective (Braithwaite et al, 2019).
The multifaceted nature of health systems and the spread of direct and indirect responsibilities across multiple sectors pose challenges for performance monitoring. In response, in recent years the World Health Organisation (WHO) (World Health Organisation [WHO], 2007) and its partners have been working to reach a broad consensus on key indicators and effective methods and measures for understanding and evaluating health systems, including inputs, processes and outcomes.
Kaplan and colleagues (2013) identify the principles of applying GST to health organisations. According to the authors, we must have: 1) patient-centred system processes. The needs and perspectives of the patient should be at the centre of the whole process in organisational terms, in terms of planning, implementation and evaluation and the involvement of professionals; 2) the creation of a system of excellence through the identification and establishment of good practices; 3) adjustments to the needs of patients, care and other circumstances; 4) considering a learning process involving scientific evidence and the experience of local change; 5) emphasising the interdependence and interconnection between processes and different actors; 6) emphasising efficient and effective teamwork, leadership and coordination; 7) considering but avoiding performance errors in the system; 8) implement rewards for elements of the system that contribute to the continuous improvement of the system; 9) encourage education/training and research for health professionals and partnerships with other relevant scientific areas (e.g. technologies); 10) promote an organisational culture of leadership and communication that strengthens teamwork and results.
- The present study involves two samples: a sample of health professionals and a sample of patients from these organizations. How did both samples select? Under which sampling technique? What’s the criterion for health professionals and patients to participate in the study? Write it clearly
With regard to the sample of professionals, all the professionals in each hospital were invited to take part in the study. The administration of each hospital sent all the professionals a link to the informed consent form and the study questionnaire via their professional email address. The participation of at least 5 per cent of the professionals in each hospital was accepted. With regard to patients, a random selection was made of those who had attended the hospital in the last 6 months, and informed consent and the link to access the questionnaire were sent by email.
- How did the hospitals select? What were the criteria for hospital selection?
The selection of hospitals was carried out by the Ministry of Health's Directorate-General for Health, which selected the three national hospitals specialising in a particular area (I can't provide this information for confidentiality reasons) in the three regions of the country (North, Centre and South).
- Provide the revised article with the complete data collection instrument (Questionnaire) for both health professionals and patients. I would like to see it. It should be in English and attached supplementary file with the revised article
The battery of instruments used with professionals is indicated in the instruments section, and they are published and validated instruments. The patient instrument is the property of the Ministry of Health, so I can't send it to you. The question used for this paper is indicated in the instruments section.
Instruments
The instrument concerning the professionals included sociodemographic questions and five scales to evaluate the variables under study: organisational culture, quality of life, psychosocial risks of work, performance management and satisfaction with work.
Organisational Culture (OC)
A translated and adapted version of the Organizational Culture Assessment In-strument (OCAI) by Cameron and Quinn (2011) was used to measure Organizational Culture. The OCAI constitutes a measure composed of 24 items, which serves to diagnose the type of organisational culture prevailing in the organisation, being organised into four dimensions with 6 items each: clan culture (α= 0.82), adhocracy culture (α= 0.85), hierarchy culture (α= 0.80) and market culture (α= 0.69).
The Portuguese version of the scale presented by Cruz and Ferreira (2015) was used and as the authors opted, in an attempt to standardise the response scales used, for a 5-point Likert type scale (1- I strongly disagree; 5 - I strongly agree).
Quality of Life (QL)
The World Health Organization Quality of Life (WHOQOL-1998) instrument was used to assess quality of life (QL).
The WHOQOL-BREF instrument comprises 26 questions (question numbers 1 and 2 on general quality of life), the answers follow a Likert type scale (from 1 to 5, the higher the score the better the quality of life). Apart from these two questions (1 and 2), the in-strument has 24 items that are organised into four dimensions: Physical QoL (7 items) (α= 0.87), Psychological QoL (6 items) (α= 0.84), Social QoL (3 items) (α= 0.64) and Envi-ronmental QoL (8 items) (α= 0.78). The Portuguese version of the WHOQOL-BREF translated and adapted by Canavarro et al. (2007) and Vaz Serra et al. (2006) was used.
Psychosocial Work Factors
The Copenhagen Psychosocial Questionnaire - COPSOQ II (middle version) by Kristensen (2002) translated and adapted into Portuguese by Silva et al. (2011) was used to measure the Psychosocial Work Factors. The instrument consists of 76 items, organised into 29 dimensions: work demands (6 items), work organisation and content (6 items), social relationships and leadership (7 items), work-individual interface 4 items), work-place values (5 items), personality (1 item), Health and well-being (7 items) and offending behaviours (4 items), responses follow a 5-point Likert-type scale (1 - never / almost never to 5 - extremely). Cronbach's Alpha values range between 0.20 in the vertical trust di-mension and 0.90 in the leadership quality dimension (Silva et al. 2011).
Performance Management (PM)
A translated and adapted version of the Performance Management Behavior Ques-tionnaire (PMBQ) by Kinicki et al. (2013) was used to measure health professionals' Performance Management (self-assessment).
The instrument has 27 items, organised into 6 dimensions: goal setting process (5 items) (α= 0.91), communication (4 items) (α= 0.86), feedback (5 items) (α= 0.85), Coaching (5 items) (α= 0.91), consequence setting (3 items) (α= 0.93) and setting and monitoring performance expectations (5 items) (α= 0.71). The response interval is a 5-point scale (1- rarely or never; 5 - often or always).
Professionals' Job Satisfaction (PJS)
A translated and adapted version of the Satisfaction of Employees in Health Care (SEHC) scale (Alpern et al., 2013) was used to assess Job Satisfaction.
The instrument was translated and adapted through the following translation and back-translation procedure: translation of the English version into Portuguese by two researchers, comparison and homogenisation of the versions, translation of the resulting version into English, comparison of this version with the original version by an expert in English, and definition of the final version with the agreement of both researchers (Fortes & Araújo, 2019). The scale consists of 20 items, 18 of them with a 4-point response scale (1- strongly disagree to 4 - strongly agree), these items with organized by three dimensions: relationship with management and supervisors (11 items) (α= 0.89), work content (5 items) (α= 0.70), relationship with co-workers (2 items) (α= 0.70). The scale also includes 2 more items: "I would recommend this service to others as a good place to work" with a 4-point response scale (1- not at all to 4 - yes, at all) and "How would you rate this health facility as a place to work on a scale from 1 (the worst) to 10 (the best)".
Patient Satisfaction (PS)
The Questionnaire on Health System User Satisfaction (QSUSS) was used to measure patient satisfaction (Department of Quality in Health/Directorate-General for Health, 2015, 2019).
The following question was used for the present study: "In your opinion, how does the Portuguese Health System work?" with the following response hypotheses: "Works well"; "Needs minor changes/adjustments"; "Needs major changes/adjustments"; "Needs to be completely restructured".
- Is there any data quality check on the collected data? If yes, then explain the procedure of the data quality check
Data cleaning was used to check for incorrect values, for example ages under 18 or over 110 (these were not identified). As the questionnaire could only be submitted with all the questions answered, no missing values were identified.
- What are the limitations and implications of the study? Write in detail
The main limitations of the study were the low level of participation by health professionals in filling in the quantitative data collection instrument, even though methodological measures were taken to increase participation, such as involving hospital administration and clinical management, internal presentation of the study in a plenary session and reinforcement of participation. This low level of participation may be due to the length of the questionnaire, which took around 30 minutes to complete, the low expectations professionals had of the impact of their participation and their relationship with the administration and management. There was a much higher response rate in organisation C, which reflected better overall results in terms of quality and results. The fact that the total number of participants was lower than expected created limitations in terms of analysing and processing the data in the reduced model proposal. Consequently, this research proposes a smaller instrument (average completion time of 10 minutes) to increase the participation rate in future research or even make it more feasible to use the model in evaluation and monitoring, in the form of a diagnostic tool for healthcare organisations.
The sample was by convenience and only included public health organisations, which limits the generalisability of the results to the universe of health organisations. It would be desirable to continue this research by applying the methodology to a random and representative sample of NHS health organisations, public health organisations with private management and private health organisations.
Implications
This study contributes to a more in-depth understanding of the factors that influence the quality of healthcare organisations and their results, and produces recommendations for healthcare organisations to meet the current challenges. The study used a systemic, multidimensional and integrative perspective to characterise and understand the factors, actors and their respective relationships and influences in SNS health organisations, at the political, organisational, human resources, professional, patient and economic-financial levels. The study used a mixed methodology that was therefore more robust, using quantitative and qualitative methods, integrating the information collected through various methods and multi-informants.
The difficulties faced by health researchers in terms of data collection are well known. The fact that this research was successful in this respect can be explained: on the one hand, because the administrations of the healthcare organisations were involved in the study from the outset, on the other hand, because there was awareness-raising among management, middle management and professionals and, finally, because the results and products of the study can have a direct impact on the assessment and monitoring of the quality and Integrated Quality and Safety System of the respective healthcare organisations.
The study involved participants from different levels of healthcare organisations, which made it possible to integrate and triangulate the information and thus understand the perception of each of the stakeholders, allowing points of agreement and discrepancies to be identified.
As a result, we have a comprehensive diagnostic model of the factors that influence results in healthcare organisations, which provides greater knowledge of the systems and the relationships between the systems and which can be used to support decision-making, planning and the implementation of improvements in NHS healthcare organisations. For the regular evaluation and monitoring of SNS health organisations, an integrated evaluation model and method was studied and proposed to assess the impact of implementing improvement measures and consequently support an evidence-based governance process. In addition to the evaluation and monitoring tools, recommendations and pointers for action are presented and discussed for the continuous and sustainable improvement of health organisations in the National Health System.
We would like to highlight some implications for practice based on the results, recommendations and products of this study: the importance and impact of organisational culture on processes and results was demonstrated; at the level of professionals, the relevance of involving all professionals was mirrored, groups with specific needs and levels were identified, and the relationship of QoL and psychosocial risks at work with other processes and results in healthcare organisations was proven; at the level of patients, there was general satisfaction with the SNS, especially with the organisations under study, however, waiting times and involvement in decision-making in relation to their health are aspects that need improvement according to patients' perceptions.
The main theoretical contribution of this work is the in-depth study of General Systems Theory in healthcare organisations. Systems and subsystems in healthcare organisations were identified, selected and studied, and the types and strength of the influences between the different systems and subsystems were analysed. The study makes it possible to verify a profile of healthcare organisation that obtained more positive indicators for all factors and all levels of analysis. Another important contribution is the relationship established between organisational culture, psychosocial risks at work and results. And the relationship between patient satisfaction and economic and financial results.
- What’s the future scope of the study? Write in detail under the context of health professionals and patients
This paper is part of a wider study on the global evaluation of healthcare in Portugal. As a result of this study, evaluation reports, recommendations and action plans have been drawn up for each of the hospitals involved and annual monitoring of the impact of continuous quality improvement will be carried out. The aim is to extend the evaluation and monitoring to other hospitals and primary healthcare organisations (health centres). Always involving the different stakeholders, administration, professionals and patients.
Reviewer 2 Report
Comment on healthcare-2555854
1. One serious mistake is that authors confuse category with specific variables. In the manuscript, you have organizational culture, performance management, psychological work factors, and economic and financial results (EFR). They are not variables, but broad categories. You need to specify them and place them at the variable level. For example, organizational cultural values include respect for people, aggressiveness, team orientation, pay attention to details, stability, innovation, and outcome-orientation. Build your hypotheses with these specific cultural values. Similarly, build your hypotheses on the dimensional level of your performance management. Each dimension can be viewed as a variable, such as goal setting, feedback, and coaching.
2. Another serious concern is the level of analysis. Organizational culture, performance management, and economic and financial results (hospital outcomes) appear to be organization-level variables, but job satisfaction and quality of life are individual-level variables. You need to clarify and justify the level of analysis before you conduct your data analysis.
3. It can be understood that hospital professionals take charge of data collection of organizational culture, performance management and other hospital-related variables. However, hospital patients do not know performance management and organizational cultural quite well.
4. The combination of professional data and patient data is a question. I wonder how you combine the professional questionnaires and patient questionnaire together. You administer their questionnaires separately, but how do you put their variables in one study? What is the total sample size? Is it 470 (professional) + 768 (patients)? You can combine their dataset of organizational culture, but how do you combine professional satisfaction with patient satisfaction?
5. Build hypotheses in the Literature and Hypotheses part. With these hypotheses, we can know what relationships receive support. Readers can cite these findings in their studies.
Keeping concise and make your ideas to the point are necessary.
Author Response
Revisions Paper
“Health Management: evaluating the relationship between or-ganizational factors, psychosocial risks at work, performance management and hospitals outcomes.”
Dear Editor and Reviewer
I greatly appreciate the careful analysis and recommendations indicated by the reviewers. I have made the requested changes; in this document, following the reviewer's recommendation, is the answer that was included in the manuscript, I indicate in track changes. I made substantial changes that greatly improved the paper, thank you very much. If you have any further questions or if needed any additional changes, please just let me know best regards Tania
1 - One serious mistake is that authors confuse category with specific variables. In the manuscript, you have organizational culture, performance management, psychological work factors, and economic and financial results (EFR). They are not variables, but broad categories. You need to specify them and place them at the variable level. For example, organizational cultural values include respect for people, aggressiveness, team orientation, pay attention to details, stability, innovation, and outcome-orientation. Build your hypotheses with these specific cultural values. Similarly, build your hypotheses on the dimensional level of your performance management. Each dimension can be viewed as a variable, such as goal setting, feedback, and coaching
In the model shown in "Figure 1 - Global health management model - final model", the initial instruments "the categories" (construct with all its dimensions) were statistically reduced until they became one-dimensional instruments that we consider "variables". The reduction procedure is shown.
The original instrument, which included the complete scales previously mentioned, consisted of a total of 241 items with an average time for completion of 30 min. In order to propose a user-friendly instrument for an effective and sustainable use in the assessment and monitoring of the management and quality of health care organizations, a process of item reduction was conducted for each of the scales under study, and a Path analysis was performed to determine the quality and robustness of the proposed model. In the item reduction process, the following analyses were conducted for each of the instruments used in the study (I.e. ; Organizational Culture Assessment Instruments, Quality of Life Assessment Instruments, Psychosocial Risks at Work Assessment Instrument, Performance Management Assessment Instrument and Job Satisfaction Assessment Instrument) 1st Exploratory Factor Analysis SPSS; 2nd Factor Reliability Analysis SPSS; 3rd Pearson Correlation Analysis SPSS; 4th Confirmatory Factor Analysis of all EQS Scales; and 5th Structural Equation Model EQS.
For the confirmatory factor analysis, structural equation modelling was performed. The ratio between the chi-square (χ2) and the degrees of freedom (gl), and the fit indices NNFI (Non-Normed Fit Index), CFI (Comparative Fit Index), RMSEA (Root-Mean-Square Error of Approximation) and interval were considered. The χ2 indicates the magnitude of the discrepancy between the observed and the modelled covariance matrix, assessing the probability of the model's fit to the data. It is generally considered its ratio in relation to the degrees of freedom (χ2/gl) whose appropriate values are between 1 and 3 (Kline, 2005, 2010). The NNFI and CFI indices calculate the relative fit of the observed model by comparing it with a base model, whose values above 0.95 indicate optimal fit and those above 0.90 indicate adequate fit (Bentler, 1990; Hu & Bentler, 1999). The RMSEA is a measure of discrepancy, being considered good results less than 0.06, but acceptable up to 0.08 (Marôco, 2014; Noronha, Pinto & Ottati, 2016;).
If the information is not sufficient or unclear, please indicate that we should clarify it. Thank you very much.
- Another serious concern is the level of analysis. Organizational culture, performance management, and economic and financial results (hospital outcomes) appear to be organization-level variables, but job satisfaction and quality of life are individual-level variables. You need to clarify and justify the level of analysis before you conduct your data analysis.
In the model presented and taking general systems theory into account, the variables quality of life and job satisfaction were considered process and outcome variables respectively, considering that, as individual-level variables, they were related to the work context and were influenced by the organisational culture and, in the case of satisfaction, also to the process variables (quality of life, psychosocial risks at work and performance).
- It can be understood that hospital professionals take charge of data collection of organizational culture, performance management and other hospital-related variables. However, hospital patients do not know performance management and organizational cultural quite well.
The battery of instruments for professionals includes instruments that assess organisational culture, performance management, etc. The instrument applied to patients only assesses their overall satisfaction with the hospital and healthcare in specific areas.
- The combination of professional data and patient data is a question. I wonder how you combine the professional questionnaires and patient questionnaire together. You administer their questionnaires separately, but how do you put their variables in one study? What is the total sample size? Is it 470 (professional) + 768 (patients)? You can combine their dataset of organizational culture, but how do you combine professional satisfaction with patient satisfaction?
The model “Figure 1. - Global health management model - final model” include data from 3 different sources: from professional (N=470), from patients (N=768) end economic and financial outcomes https://benchmarking-acss.min-saude.pt/BH_AcessoDashboard
For each of the hospitals, the information was collected, and the total database included the results per subject in the case of professionals, and in the case of patients and economic-financial outcomes, the average value of the results for each of the hospitals.
Patients were not paired with professionals, the cluster analysed was the hospital, the total results of professionals for each of the variables, the total results of the hospital for patient satisfaction and the total result of economic and financial results.
- Build hypotheses in the Literature and Hypotheses part. With these hypotheses, we can know what relationships receive support. Readers can cite these findings in their studies.
(Included at the end of introduction)
Based on the established objective, the following research hypotheses are proposed:
H1 - A dominant and well-defined organisational culture has a positive influence on the results of public health organisations;
H2 - A high quality of life among health professionals has a positive influence on the results of public health organisations;
H3 - Greater psychosocial risks at work have a negative influence on the results of public health organisations;
H4 - More effective performance management has a positive influence on the results of public health organisations;
H5 - A dominant and well-defined organisational culture is associated with a higher quality of life for health professionals;
H6 - A dominant and well-defined organisational culture is associated with fewer psychosocial risks at work.
H7 - A dominant and well-defined organisational culture is associated with better performance management among healthcare professionals.
(Included in the discussion)
These results provide a direct answer to the study hypotheses.
H1 - A dominant and well-defined organisational culture has a positive influence on the results of public health organisations. The results confirm statistically significant correlations (P>0.05) between the types of organisational culture and the results under analysis (staff satisfaction, patient satisfaction and economic and financial performance results), highlighting the strong correlation between clan culture and staff satisfaction (R=0.67).
H2 - High quality of life among health professionals has a positive influence on the results of public health organisations. The results partially confirm this hypothesis. There is a moderate and statistically significant correlation (R=0.31) between professionals' quality of life and job satisfaction.
H3 - Greater psychosocial risks at work have a negative influence on the results of public health organisations. The results confirm this hypothesis. There is a strong and statistically significant negative correlation (R=-0.75) between psychosocial risks at work and job satisfaction. The relationship is statistically significant, but weak in relation to patient satisfaction and economic and financial performance results.
H4 - Effective performance management has a positive influence on the results of public health organisations. The results partially confirm this hypothesis. There is a strong and statistically significant correlation (R=0.35) between performance management and job satisfaction. The relationship is not confirmed in relation to patient satisfaction and economic and financial performance results.
H5 - A dominant and well-defined organisational culture is associated with a higher quality of life for healthcare professionals. Given this hypothesis, we found that quality of life is positively and statistically significantly correlated with Organisational Culture (R=0.22).
H6 - A dominant and well-defined organisational culture is associated with fewer psychosocial risks at work. In view of this hypothesis, we found that psychosocial risks at work are strongly negatively and statistically significantly correlated with organisational culture (R=-0.61).
H7 - A dominant and well-defined organisational culture is associated with better performance management among healthcare professionals. Given this hypothesis, we found that performance management is positively and statistically significantly correlated with organisational culture (R= 0.15).
Healthcare organisations face a number of challenges in terms of the results they set out to achieve. Health and well-being, as well as professional and patient satisfaction, will influence economic and financial results (EU-OSHA, 2009; OECD, 2018, 2020a, 2020b). Psychosocial risk factors can be individual factors, organisational factors and the interaction between them, the level of exposure to risks and the concrete impact it has on the professional's general health. Individual characteristics include psychological factors such as coping style, personality and cognitive characteristics and sociodemographic characteristics such as gender, age, marital status and physical and mental health history. EU-OSHA (2007, 2012) and the Directorate-General for Health (2015) describe the main factors that should be considered when assessing psychosocial risks at work: the organisation's culture and leadership relationships, work content, work overload, working hours, level of control and autonomy, interpersonal relationships at work, career development possibilities, work-family relationships, and work environment and equipment. The impact of psychosocial risks on the health and performance of professionals can have chronic and long-term consequences (OECD, 2018, 2020a, 2020b), in terms of their physical health, mental health and chronic stress and absenteeism from work (Gaspar et al, 2023).
The model obtained can be understood in the light of the General System Theory, i.e. the interrelationship and influence between the variables makes it possible to identify and understand factors in the organisation, the professionals and the results and their relationship that influence the quality of life of the professionals, the performance and the results of the organisation (Berghofer et al., 2020; BNQ, 2018; Gaspar, 2020).
The OECD (2017) and Clarkson et al. (2018) argue that assessing the quality of healthcare organisations from a systemic perspective must take into account various fundamental concepts, such as leadership, strategy, plans, patients, society, information and knowledge, people, processes and outcomes.
Reviewer 3 Report
This is obviously a very important topic, and the authors are to be commended for undertaking such an ambitious project. My main suggestion is to add a common-sense interpretation of the results so that they are more accessible to a broad readership. What are psychosocial risks--burnout? staff turnover? What is a well-defined organizational culture? Can an authoritarian culture be well-defined? What are the various types of culture? Does it matter what the ratio of supervisors to staff is? What is performance management? Is patient satisfaction a good indicator of effective medical treatment? What about prescribing excessive opiates? Can patients really judge effective treatment? How about measures of hospital readmission or other such indicators? What are desirable financial outcomes--spending a lot of money or being more economical? I can see problems arising from not buying expensive equipment or from waste. How can a hospital manager profit from these results? What do they tell the lay public about how hospitals can be improved? How do these results suggest applying practices that are not in general use? Yes, lots of factors must be taken into account to improve care, but what are the practical recommendations? Try to go beyond confirming obviously desirable practices to new findings. Separate the wheat from the chaff.
Writing suggestions:]
line 33--omit and creating value
57f--ideal not adequate
72--omit hyphen
79--omit second comma
81--enhance, not increases
83--omit On the other hand
85--omit commas
89--which is related to
104--occupational what?
106ff--break up sentence
149--quality of what?
171--colon after samples
176--omit second comma
178--were not was
190--semicolon not comma
200--explain adhocracy
208f--reword
265--semicolon instead of second comma
266--use parentheses
282--describe telephone interview procedure--sampling: cell phones too? rate of refusal?
313ff--unclear
318f--reword
344--there before resulted
345--was after very
346--start new sentence with It...for not of
353--using the Lagrange
359--start new sentence with It
368--with the Lagrange
371--omit From
384--by the Lagrange
402--semicolon not comma
410--evaluation of
422--by the Lagrange
449--semicolon not comma
452--comma after test
481--apostrophe
527--it refers to what?
570--incomplete sentence
Use shorter sentences where possible. Use and before the last item in a series. Use more semicolons.
Author Response
Revisions Paper
“Health Management: evaluating the relationship between or-ganizational factors, psychosocial risks at work, performance management and hospitals outcomes.”
Dear Editor and Reviewer
I greatly appreciate the careful analysis and recommendations indicated by the reviewers. I have made the requested changes; in this document, following the reviewer's recommendation, is the answer that was included in the manuscript, I indicate in track changes. I made substantial changes that greatly improved the paper, thank you very much. If you have any further questions or if needed any additional changes, please just let me know best regards Tania
This is obviously a very important topic, and the authors are to be commended for undertaking such an ambitious project. My main suggestion is to add a common-sense interpretation of the results so that they are more accessible to a broad readership.
What are psychosocial risks--burnout? staff turnover? What is a well-defined organizational culture? Can an authoritarian culture be well-defined? What are the various types of culture? Does it matter what the ratio of supervisors to staff is? What is performance management? Is patient satisfaction a good indicator of effective medical treatment? What about prescribing excessive opiates? Can patients really judge effective treatment? How about measures of hospital readmission or other such indicators? What are desirable financial outcomes--spending a lot of money or being more economical? I can see problems arising from not buying expensive equipment or from waste. How can a hospital manager profit from these results? What do they tell the lay public about how hospitals can be improved? How do these results suggest applying practices that are not in general use? Yes, lots of factors must be taken into account to improve care, but what are the practical recommendations? Try to go beyond confirming obviously desirable practices to new findings. Separate the wheat from the chaff.
Information included in Introduction
General Systems Theory applied to Healthcare organizations
General Systems Theory (GST) is considered a holistic, interdisciplinary approach to science that emphasises the importance of context and environment. A system can be defined as the interaction and/or interdependence of various parts that form the whole, the various parts as a whole serving a common goal and interacting with the environment external to the system (Cordon, 2013). TGS focuses on the relationships between the parts and how they articulate as a whole. The properties of a system are determined by how the parts are organised and interact. In this way, a system is a complex, interconnected structure of parts. The whole is different and even greater than the sum of all the parts (Chikere & Nwoka, 2015).
Health systems are open complex systems with interrelated subsystems that function in a dynamic equilibrium managed by communication and information (Chuang & Inder, 2009) and consequently must be understood from a systemic perspective (Braithwaite et al, 2019).
The multifaceted nature of health systems and the spread of direct and indirect responsibilities across multiple sectors pose challenges for performance monitoring. In response, in recent years the World Health Organisation (WHO) (World Health Organ-isation [WHO], 2007) and its partners have been working to reach a broad consensus on key indicators and effective methods and measures for understanding and evaluating health systems, including inputs, processes and outcomes.
Kaplan and colleagues (2013) identify the principles of applying GST to health or-ganisations. According to the authors, we must have: 1) patient-centred system processes. The needs and perspectives of the patient should be at the centre of the whole process in organisational terms, in terms of planning, implementation and evaluation and the in-volvement of professionals; 2) the creation of a system of excellence through the identi-fication and establishment of good practices; 3) adjustments to the needs of patients, care and other circumstances; 4) considering a learning process involving scientific evidence and the experience of local change; 5) emphasising the interdependence and intercon-nection between processes and different actors; 6) emphasising efficient and effective teamwork, leadership and coordination; 7) considering but avoiding performance errors in the system; 8) implement rewards for elements of the system that contribute to the continuous improvement of the system; 9) encourage education/training and research for health professionals and partnerships with other relevant scientific areas (e.g. technolo-gies); 10) promote an organisational culture of leadership and communication that strengthens teamwork and results.
The evaluation of healthcare organisations is carried out in a segmented way: or-ganisational factors are evaluated, or psychosocial risks at work, or the performance of professionals, or the satisfaction of professionals or patients; no model has been identified in the literature that evaluates all these aspects of the organisation at the same time and that makes it possible to establish the relationship between them.
In view of the above, the central objective of this study is to propose a comprehensive and multidimensional model to assess the management and quality of public health organisations, which integrates and relates different component processes of the system, namely, the impact of organisational (OC), psychosocial (RPT), professional-related (QL, performance management) factors, and illustrates how they relate to outcomes (patient satisfaction, professional satisfaction and economic and financial performance outcomes).
Based on the established objective, the following research hypotheses are proposed:
H1 - A dominant and well-defined organisational culture has a positive influence on the results of public health organisations;
H2 - A high quality of life among health professionals has a positive influence on the results of public health organisations;
H3 - Greater psychosocial risks at work have a negative influence on the results of public health organisations;
H4 - More effective performance management has a positive influence on the results of public health organisations;
H5 - A dominant and well-defined organisational culture is associated with a higher quality of life for health professionals;
H6 - A dominant and well-defined organisational culture is associated with fewer psychosocial risks at work.
H7 - A dominant and well-defined organisational culture is associated with better performance management among healthcare professionals.
Information included in discussion
These results provide a direct answer to the study hypotheses.
H1 - A dominant and well-defined organisational culture has a positive influence on the results of public health organisations. The results confirm statistically significant correlations (P>0.05) between the types of organisational culture and the results under analysis (staff satisfaction, patient satisfaction and economic and financial performance results), highlighting the strong correlation between clan culture and staff satisfaction (R=0.67).
H2 - High quality of life among health professionals has a positive influence on the results of public health organisations. The results partially confirm this hypothesis. There is a moderate and statistically significant correlation (R=0.31) between professionals' quality of life and job satisfaction.
H3 - Greater psychosocial risks at work have a negative influence on the results of public health organisations. The results confirm this hypothesis. There is a strong and statistically significant negative correlation (R=-0.75) between psychosocial risks at work and job satisfaction. The relationship is statistically significant, but weak in relation to patient satisfaction and economic and financial performance results.
H4 - Effective performance management has a positive influence on the results of public health organisations. The results partially confirm this hypothesis. There is a strong and statistically significant correlation (R=0.35) between performance management and job satisfaction. The relationship is not confirmed in relation to patient satisfaction and economic and financial performance results.
H5 - A dominant and well-defined organisational culture is associated with a higher quality of life for healthcare professionals. Given this hypothesis, we found that quality of life is positively and statistically significantly correlated with Organisational Culture (R=0.22).
H6 - A dominant and well-defined organisational culture is associated with fewer psychosocial risks at work. In view of this hypothesis, we found that psychosocial risks at work are strongly negatively and statistically significantly correlated with organisational culture (R=-0.61).
H7 - A dominant and well-defined organisational culture is associated with better performance management among healthcare professionals. Given this hypothesis, we found that performance management is positively and statistically significantly correlated with organisational culture (R= 0.15).
Healthcare organisations face a number of challenges in terms of the results they set out to achieve. Health and well-being, as well as professional and patient satisfaction, will influence economic and financial results (EU-OSHA, 2009; OECD, 2018, 2020a, 2020b). Psychosocial risk factors can be individual factors, organisational factors and the interaction between them, the level of exposure to risks and the concrete impact it has on the professional's general health. Individual characteristics include psychological factors such as coping style, personality and cognitive characteristics and sociodemographic characteristics such as gender, age, marital status and physical and mental health history. EU-OSHA (2007, 2012) and the Directorate-General for Health (2015) describe the main factors that should be considered when assessing psychosocial risks at work: the organisation's culture and leadership relationships, work content, work overload, working hours, level of control and autonomy, interpersonal relationships at work, career development possibilities, work-family relationships, and work environment and equipment. The impact of psychosocial risks on the health and performance of professionals can have chronic and long-term consequences (OECD, 2018, 2020a, 2020b), in terms of their physical health, mental health and chronic stress and absenteeism from work (Gaspar et al, 2023).
The model obtained can be understood in the light of the General System Theory, i.e. the interrelationship and influence between the variables makes it possible to identify and understand factors in the organisation, the professionals and the results and their relationship that influence the quality of life of the professionals, the performance and the results of the organisation (Berghofer et al., 2020; BNQ, 2018; Gaspar, 2020).
The OECD (2017) and Clarkson et al. (2018) argue that assessing the quality of healthcare organisations from a systemic perspective must take into account various fundamental concepts, such as leadership, strategy, plans, patients, society, information and knowledge, people, processes and outcomes.
AND
This study contributes to a more in-depth understanding of the factors that influence the quality of healthcare organisations and their results, and produces recommendations for healthcare organisations to meet the current challenges. The study used a systemic, multidimensional and integrative perspective to characterise and understand the factors, actors and their respective relationships and influences in SNS health organisations, at the political, organisational, human resources, professional, patient and economic-financial levels. The study used a mixed methodology that was therefore more robust, using quantitative and qualitative methods, integrating the information collected through various methods and multi-informants.
The difficulties faced by health researchers in terms of data collection are well known. The fact that this research was successful in this respect can be explained: on the one hand, because the administrations of the healthcare organisations were involved in the study from the outset, on the other hand, because there was awareness-raising among man-agement, middle management and professionals and, finally, because the results and products of the study can have a direct impact on the assessment and monitoring of the quality and Integrated Quality and Safety System of the respective healthcare organisa-tions.
The study involved participants from different levels of healthcare organisations, which made it possible to integrate and triangulate the information and thus understand the perception of each of the stakeholders, allowing points of agreement and discrepancies to be identified.
As a result, we have a comprehensive diagnostic model of the factors that influence results in healthcare organisations, which provides greater knowledge of the systems and the relationships between the systems and which can be used to support decision-making, planning and the implementation of improvements in NHS healthcare organisations. For the regular evaluation and monitoring of SNS health organisations, an integrated eval-uation model and method was studied and proposed to assess the impact of implementing improvement measures and consequently support an evidence-based governance pro-cess. In addition to the evaluation and monitoring tools, recommendations and pointers for action are presented and discussed for the continuous and sustainable improvement of health organisations in the National Health System.
We would like to highlight some implications for practice based on the results, recommendations and products of this study: the importance and impact of organisational culture on processes and results was demonstrated; at the level of professionals, the relevance of involving all professionals was mirrored, groups with specific needs and levels were identified, and the relationship of QoL and psychosocial risks at work with other processes and results in healthcare organisations was proven; at the level of patients, there was general satisfaction with the SNS, especially with the organisations under study, however, waiting times and involvement in decision-making in relation to their health are aspects that need improvement according to patients' perceptions.
The main theoretical contribution of this work is the in-depth study of General Systems Theory in healthcare organisations. Systems and subsystems in healthcare or-ganisations were identified, selected and studied, and the types and strength of the influences between the different systems and subsystems were analysed. The study makes it possible to verify a profile of healthcare organisation that obtained more positive indicators for all factors and all levels of analysis. Another important contribution is the relationship established between organisational culture, psychosocial risks at work and results. And the relationship between patient satisfaction and economic and financial results.
Information included in conclusions
The main limitations of the study were the low level of participation by health professionals in filling in the quantitative data collection instrument, even though methodological measures were taken to increase participation, such as involving hospital administration and clinical management, internal presentation of the study in a plenary session and reinforcement of participation. This low level of participation may be due to the length of the questionnaire, which took around 30 minutes to complete, the low expectations professionals had of the impact of their participation and their relationship with the administration and management. There was a much higher response rate in organisation C, which reflected better overall results in terms of quality and results. The fact that the total number of participants was lower than expected created limitations in terms of analysing and processing the data in the reduced model proposal. Consequently, this research proposes a smaller instrument (average completion time of 10 minutes) to increase the participation rate in future research or even make it more feasible to use the model in evaluation and monitoring, in the form of a diagnostic tool for healthcare or-ganisations.
The sample was by convenience and only included public health organisations, which limits the generalisability of the results to the universe of health organisations. It would be desirable to continue this research by applying the methodology to a random and representative sample of NHS health organisations, public health organisations with private management and private health organisations.
AND
This paper is part of a wider study on the global evaluation of healthcare in Portugal. As a result of this study, evaluation reports, recommendations and action plans have been drawn up for each of the hospitals involved and annual monitoring of the impact of continuous quality improvement will be carried out. The aim is to extend the evaluation and monitoring to other hospitals and primary healthcare organisations (health centres). Always involving the different stakeholders, administration, professionals and patients.
The Portuguese health system, the NHS and the basic health law (Law no. 48/1990, 1990; Law no. 95/2019, 2019) present principles and structure that fit into Systems Theory and allow for the understanding, assessment and improvement of Health Systems at the management, quality and performance levels. From this perspective, for a better un-derstanding of a complex and multidimensional phenomenon, such as the functioning of Portuguese health organizations, the following factors should be included: organizational factors, professional factors, patient factors, as well as social, cultural, economic, and po-litical factors (Gaspar, 2020).
The following are the main recommendations for managers of NHS health organi-zations: (a) To periodically assess and monitor the quality and performance of health organizations from a systemic and integrative perspective (integrating inputs, processes and outcomes); (b) Promote Healthy Workplaces by improving psychosocial working conditions, promoting physical, social and mental health of professionals, for instance by providing them with more availability for non-care activities such as management skills, research and training.
All the corrections below have been inserted into the document.
line 33--omit and creating value
57f--ideal not adequate
72--omit hyphen
79--omit second comma
81--enhance, not increases
83--omit On the other hand
85--omit commas
89--which is related to
104--occupational what?
106ff--break up sentence
149--quality of what?
171--colon after samples
176--omit second comma
178--were not was
190--semicolon not comma
200--explain adhocracy
Adocracy culture makes the organisation a dynamic, entrepreneurial and creative place to work. Professionals are bold and take risks. Leaders are considered innovative, entrepreneurial, visionary and risk-taking. The organisation is united by a commitment to experimentation and innovation. The emphasis is on being ahead, and in the long term, the organisation aims for growth and the acquisition of new resources. Success means winning unique and new products and services. Being a product or service leader is valued. The organisation encourages individual initiative, autonomy and freedom.
208f--reword
265--semicolon instead of second comma
266--use parentheses
282--describe telephone interview procedure--sampling: cell phones too? rate of refusal?
The patients were contacted by telephone by professionals from the respective hospital. The professionals were trained by the research team and followed a script. The telephone interview initially contained information on informed consent, and if the patient agreed to take part, the professional began by asking questions related to satisfaction with healthcare.
313ff--unclear
318f--reword
344--there before resulted
345--was after very
346--start new sentence with It...for not of
353--using the Lagrange
359--start new sentence with It
368--with the Lagrange
371--omit From
384--by the Lagrange
402--semicolon not comma
410--evaluation of
422--by the Lagrange
449--semicolon not comma
452--comma after test
481--apostrophe
527--it refers to what?
570--incomplete sentence
Use shorter sentences where possible. Use and before the last item in a series. Use more semicolons.

Round 2
Reviewer 1 Report
NA
MOderate English editing is required